# Measuring Density and Similarity of Task Relevant Information in Neural Representations

## Abstract

Neural models achieve state-of-the-art performance due to their ability to extract salient features useful to downstream tasks. However, our understanding of how this task-relevant information is included in these networks is still incomplete. In this paper, we examine two questions (1) how densely is information included in extracted representations, and (2) how similar is the encoding of relevant information between related tasks. We propose metrics to measure information density and cross-task similarity, and perform an extensive analysis in the domain of natural language processing, using four varieties of sentence representation and 13 tasks. We also demonstrate how the proposed analysis tools can find immediate use in choosing tasks for transfer learning.

## 1 Introduction

Neural models achieve state of the art performance on a wide range of tasks in computer vision, machine learning and natural language processing (Goodfellow et al., 2016). The key to the success of these models is their ability to automatically create rich representations of the input (i.e images, words or sentences) that possess the salient information necessary to perform particular tasks.

However, despite the success of these neural feature extractors, most neural representations are hard to interpret (Shi et al., 2016b; Lipton, 2016). Recent research has thus attempted to answer the natural question: *what* information about the input do neural representations capture? For example, in natural language processing (NLP) tasks – the main test bed upon which we examine the methods in this paper – Shi et al. (2016a;b) reveal that syntactic phenomena are captured by neural encoder-decoder translation systems, Adi et al. (2016) demonstrate that sentence representations are aware of sentence structure like length, word-order and word-content, and Conneau et al. (2018) probe sentence representations for 10 linguistic properties.

However, while previous work has given us an idea of *what* information is contained, it is still unclear *how* this information is captured. In this paper, we attempt to fill this gap in our knowledge of interpreting neural representations, specifically focusing on the following questions:

- How densely is the information encoded within the representation, and across what dimensions is it distributed?
- How is the information shared, for a given input, between two similar (or dissimilar) tasks?

We examine these questions through a comprehensive quantitative study of how popular sentence representations encode information. We propose methods to characterize the *information density* – how densely information necessary to solve a downstream task is encoded (§3) – and *information similarity* – how similarly information is encoded for different tasks (§4). For information density, we introduce the notion of "Condensed Feature Sets" (CFS), a minimal set of features required for a classifier to achieve a certain level of accuracy on a target task. For information similarity, we take a CFS for a given task and compare it to the CFS for other tasks, which provides cues about how many common features are shared among them.

We further demonstrate that these metrics, in addition to satisfying our curiosity, have immediate practical use; specifically allowing us to choose tasks for transfer learning (§5). Transfer learning

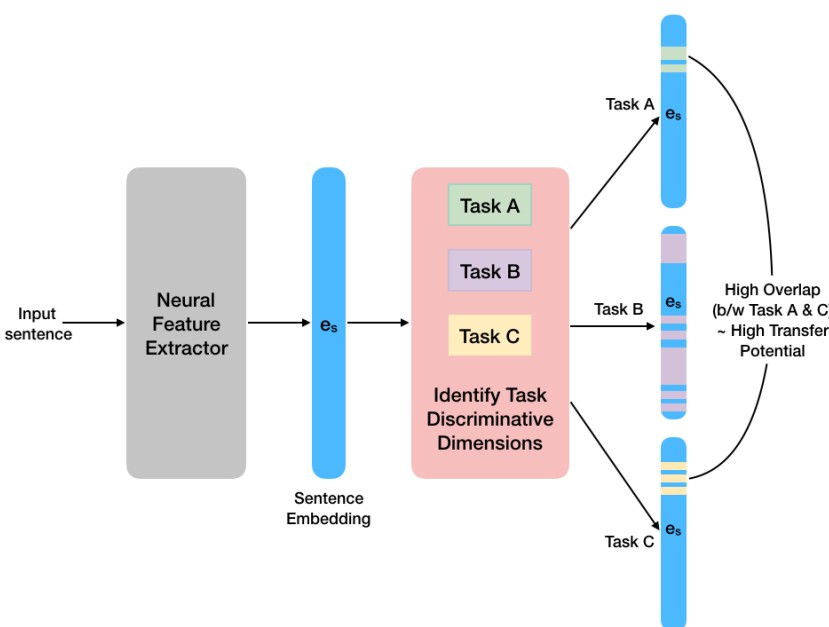

Figure 1: Input representation from a neural extractor is investigated for task-specific information. Here, information for Task A and C is compactly encoded, whereas information for Task B is dispersed. Also, similar dimensions in Task A and C are discriminative and hence the tasks require similar information and form good candidates for transfer learning.

is the act of training on one task then transferring the knowledge acquired therefrom to improve performance on another task, and is most effective when these tasks are highly related (Pan & Yang, 2010). We directly use our information similarity metric between two tasks as a proxy for transfer learning potential between them.

In experiments (§6), we examine these metrics over 4 different neural sentence encoders and 13 diverse syntactic or semantic natural language tasks. Using our measures of information density, we discover that information necessary for many of these tasks is concentrated in a very few units (§7.1). Further, information similarity metrics are able to predict good candidate tasks for transfer learning that correlate highly with actual transfer learning performance (§7.2).

## 2 CONCEPTUAL OVERVIEW

As discussed in the introduction, this paper delineates two concepts: "information density" and "information similarity". In this section, we give an informal and intuitive explanation of these concepts and their utility, while in the following sections (§3 and §4) we provide mathematical details to instantiate these concepts explicitly.

Information density is defined with respect to a particular method of encoding inputs into hidden representations, and a particular down-stream task that uses these representations. We treat the encoding of this task-specific information as "dense" if the necessary information to solve the task is largely encoded in only a few elements of the hidden representation, and "dispersed" if many neurons play a significant role in solving the tasks. This equips us to ask questions such as "Is the information for task A encoded more compactly than the information for task B?" (which can be answered in affirmative for the case in Figure 1). One can also answer questions like "Is encoder X or encoder Y able to encode the information for task A in a more compact way?"

Information similarity is defined with respect to two tasks in the context of a common fixed encoder. Specifically, we define two tasks as similar if similar elements in the hidden representation contribute to solving the two tasks. This enables us to inquire "Is task A more similar to task B or task C?" (For the case in figure 1, the answer to the question is Task C). This allows us to empirically answer

questions about which tasks are most related to each other from the point of view of a particular encoded representation, which has practical applications in transfer learning (as shown in §5).

# 3 METRICS FOR INFORMATION DENSITY

More formally, to measure information density, let $\mathbf{X} = [\boldsymbol{x}_1, \boldsymbol{x}_2, \ldots, \boldsymbol{x}_n]^{\mathsf{T}}$ be input representations, where $\boldsymbol{x}_i \in \mathbb{R}^d$. These input representations capture (to a varying degree) information essential to solve a set of classification tasks $\mathcal{T}$. Each task $t \in \mathcal{T}$ comprises of a set of $\langle \boldsymbol{x}, y \rangle$ pairs. To quantify the information in the input representations $\mathbf{X}$, about a specific classification task $t$, we measure the accuracy a classifier $f \in \mathcal{F}$ would obtain on the task $t$ using $\mathbf{X}$. We restrict $\mathcal{F}$ to be a set of linear functions over the input space $\mathbf{X}$.

If instead of using all the $d$ dimensions, we require only a handful of $k \ll d$ dimensions to retain a large fraction of the accuracy, then we can conclude that the information density for task $t$ is high for the input representations $\mathbf{X}$. To formalize this, we first define accuracy *score* of the best classifier $f$ for task $t$ using only a subset $\gamma$ of the columns of $\mathbf{X}$,

$$score(\mathcal{F}, t, \gamma) = \max_{f \in \mathcal{F}} \frac{\sum_{\langle \boldsymbol{x}_i, y_i \rangle \in t} \left( \mathbb{1}(f(\boldsymbol{x}_{i\gamma}) = y_i) \right)}{n}$$
$$\text{where:} \quad \boldsymbol{x}_{i\gamma} = [x_{ij} \mid j \in \gamma]$$

We then define the concept of a Condensed Feature Set (CFS). Let $\text{CFS}_\alpha(\mathcal{F}, T) \subseteq \{1, 2, \ldots, d\}$ be the *smallest* subset such that:

$$score(\mathcal{F}, t, \text{CFS}) \geq \alpha(score(\mathcal{F}, t, \{1, 2, \ldots, d\}) - b_t) + b_t$$

where $b_t$ is the base rate (i.e the percentage of examples in the majority class of task $t$), and $\alpha$ is a hyperparameter that we dub the 'retention ratio'. Intuitively, CFS is the smallest subset of dimensions that are able to retain $\alpha$ times the original accuracy[1] gain over the majority class baseline. It is worthwhile to note that the CFS depends on the choice of representation. Two different ways to represent input could lead to altogether different CFS for the same task.

Computing CFS is a combinatorially hard problem, and we use a greedy forward selection approach to approximate it. We start with an empty set $C$ and iteratively add dimension $i \in \{1, 2, \ldots, d\} \setminus C$ that provides the highest increase in classification accuracy until the set $C$ satisfies the criterion for it to be the CFS set. It is important to note that forward selection provides an upper bound on the cardinality of CFS for a given task, classifier and a representation choice. However, empirically we find that even this approximate method results in quite small sets of salient features in our experiments in Section 7. Also, notably the CFS does not imply that the other dimensions are devoid of any information, but rather that task specific information is sufficiently contained in the CFS.

# 4 METRICS FOR INFORMATION SIMILARITY

In this section, we tackle the problem of evaluating information similarity between two tasks by using either the previously described CFS (§3), or direct comparisons between classifier weights.

## 4.1 ESTIMATING INFORMATION SIMILARITY USING CFS

To assess the information similarity for two tasks $t_1, t_2$, we first compute their CFSs $C_1$ and $C_2$, then compute a metric of similarity between them.

A first intuitive attempt at a metric is **set overlap** ($\frac{|C_1 \cap C_2|}{|C_1 \cup C_2|}$). While the fraction of common hidden units in the condensed feature sets is a straightforward metric, it has two key drawbacks. First, set overlap does not take into account the relative influence of each dimension on classification

---

[1]Original accuracy is the accuracy score attained using the full set of dimensions

performance. While this could potentially be addressed by some sort of weighting of dimensions, a more concerning drawback is that there could be a high redundancy (or correlation) among different hidden units leading to the possibility of two sets with very small overlap potentially encoding very similar information. These drawbacks preclude us from using this metric.

Instead, we propose a second metric, **information transfer**. We wish to evaluate how useful the information content in $C_1$ is to the task $t_2$. One way to approximate this is to compute the accuracy of a classifier on task $t_2$ using the features $C_1$. This metric is robust to the redundancy problem mentioned earlier since two redundant sets would yield very similar accuracies using a linear classifier.

## 4.2 Estimating Information Similarity using Classifier Weights

An alternative way to measure the information similarity for two tasks is to compute the similarity of the classifier weights trained for the two tasks. Specifically, using the representation $X$, we train two linear classifiers and then compute the inverse distance between the classifier weights. Let $\mathbf{w}_1$ and $\mathbf{w}_2$ be the normalized absolute classifier weights, such that they are positive and sum up to 1, then we can define $similarity(t_1, t_2) = \|\mathbf{w}_1 - \mathbf{w}_2\|^{-1}$.

By using the absolute values of the learned weights, our notion of information similarity does not suffer from cases where the tasks have a strong negative correlation. In the extreme case of perfect negative correlation, the learned vectors would be $\mathbf{w}$ and $-\mathbf{w}$, and we would rightly estimate a high information similarity between two clearly related tasks.

## 5 Application to Transfer Learning

We can use our formalization of condensed feature sets and information similarity to solve a common problem in transfer learning – of choosing appropriate tasks for transfer learning. More formally, we have a set $\mathcal{T}$ of classification tasks and a representation $\mathbf{X}$, consider the following problem:

> Given a primary task $t_p \in \mathcal{T}$, find a candidate task $t_c \in \{\mathcal{T} \setminus t_p\}$ such that transfer learning from $t_c$ would lead to the highest performance gain on the task $t_p$.

This is a common problem when the primary task $t_p$ is a low resource task and there are potential high resource candidate tasks which could be used for learning from. The most common learning methodology is to train the candidate task first, and then transfer it's sentence encoder weights to the main task by fine-tuning (Mou et al., 2016; Conneau et al., 2017; 2018). This problem of finding the best candidate task for a given main task, $t_p$, can be trivially solved by trial and error. However, this requires training $2n$ neural networks (for each task, once to train the candidate task and other to fine tune after pre-training), where $n$ is the number of candidate tasks ($\mathcal{T} \setminus t_p$). Training such a large number of networks can be computationally expensive, and to the best of our knowledge, there are no efficient solutions to this problem. Practitioners often rely on their intuitive understanding of the similarity of tasks as a proxy to the task's transfer potential or transferability. Using our transfer similarity criterion, we can produce a ranked recommendation list of candidate tasks.

## 6 Experimental Setup

In this work, we examine our methods on popular natural language processing tasks using four different choices of sentence representations (a brief overview is presented in Table 1).

**Task Details**: To determine task-specific information density in sentence representations, and calculate and evaluate predicted transfer potentials, we design a set of experiments on a diverse suite of text classification tasks. The tasks can be divided into:

*High Resource Tasks* include textual entailment on Stanford Natural Language Inference (SNLI) dataset, sentiment analysis (on IMDB and binary version of Stanford Sentiment Treebank, SST (b)), Quora question deduplication task (QUORA) and probing tasks from Conneau & Kiela (2018) comprising of bigram shift (BShift) which tests if a bigram in a sentence is inverted, Tense task which asks for the tense of the main-clause verb, sentence length (SentLen) classifies a sentence into equal-width bins according to number of words in a sentence, and finally, Semantic Odd Man

| Name | Dimensions | Objective/Description |
|---|---|---|
| SkipThought (Kiros et al., 2015) | 4800 | Encode a sentence to predict the sentences around it |
| SIF (Arora et al., 2016) | 300 | Weighted average of word vectors and then modify a bit using PCA/SVD |
| InferSent (Williams et al., 2017) | 4096 | Universal sentence representations trained on supervised NLI task |
| ParaNMT (Wieting & Gimpel, 2017) | 4096 | Train paraphrastic sentence embeddings on English-English sentential paraphrase pairs |

Table 1: A brief overview of the sentence representations we experiment with in this work

Out (SOMO) tests whether a noun or a verb is out of context (replaced) in a sentence.

*Low Resource Tasks* include textual entailment on sentences involving compositional knowledge (SICK corpus), paraphrase detection (on Microsoft Research Paraphrase Corpus (MRPC)), question classification (TREC) and the fine counterpart of sentiment analysis on Stanford Sentiment Treebank (SST (f)), which includes five sentiment classes.

Since, we study the transfer from high resource tasks on low resource tasks, we control for varying dataset sizes of high resource tasks, and clip all the high resource tasks to 60K sentences. Further all the low resource tasks contain only less than 10K sentences. In our test suite, there are 4 tasks whose inputs are 2 sentences, including SNLI, SICK, MRPC, QUORA.

**Implementation Details**: To measure the task-specific information density (§3), we compute CFS (with $\alpha = 0.8$) for each task for different choices of sentence representations. We use the greedy forward selection and a logistic regression classifier with $L2$ penalty and a regularization constant $C = 1.0$. For 2-sentence input tasks, we use a bilinear logistic regression model to capture interactions between different features of each sentences.

To calculate information similarity (§4), we use the both the information transfer metric (§4.1), and classifier weight difference (§4.2). Classifier weight difference metric is only applicable in cases where the number of features between the tasks are of the same size. Thus, 2 sentence input tasks and 1 sentence input cannot be compared using the metric. However, CFS based metric doesn't suffer from such a problem.

To validate our transfer potentials, we train the four low resouce tasks (SST (f), TREC, MRPC, and SICK) by first pre-training on the other candidate tasks and subsequently fine-tuning. All models contain the same sentence encoder[2] and a logistic output layer. The tasks having two input sentences (SICK, SNLI, MRPC, QUORA) share the sentence encoder. The vocabulary is shared across the tasks, and is a union of individual task vocabularies.

# 7    RESULTS AND DISCUSSION

## 7.1    RESULTS ON INFORMATION DENSITY

We compute the Condensed Feature Sets (CFS) for the above mentioned tasks on four different sentence representations. We plot our findings in Figure 2. Clearly, we require only a few dimensions to capture the majority of the task information. For instance, in all the four representations at least 8 out of 10 tasks attain a relative accuracy of 90%, by merely using less than equal to 10 dimensions.

Interestingly, SST (b) (2 classes) is more densely represented than SST (f) (5 classes) owing to the difference in granularity of information content. Also, probing tasks are generally very densely captured as they are designed to verify presence of simple linguistic properties in sentence embeddings.

---

[2] embedding layer (H=128) followed by LSTM (H=128)

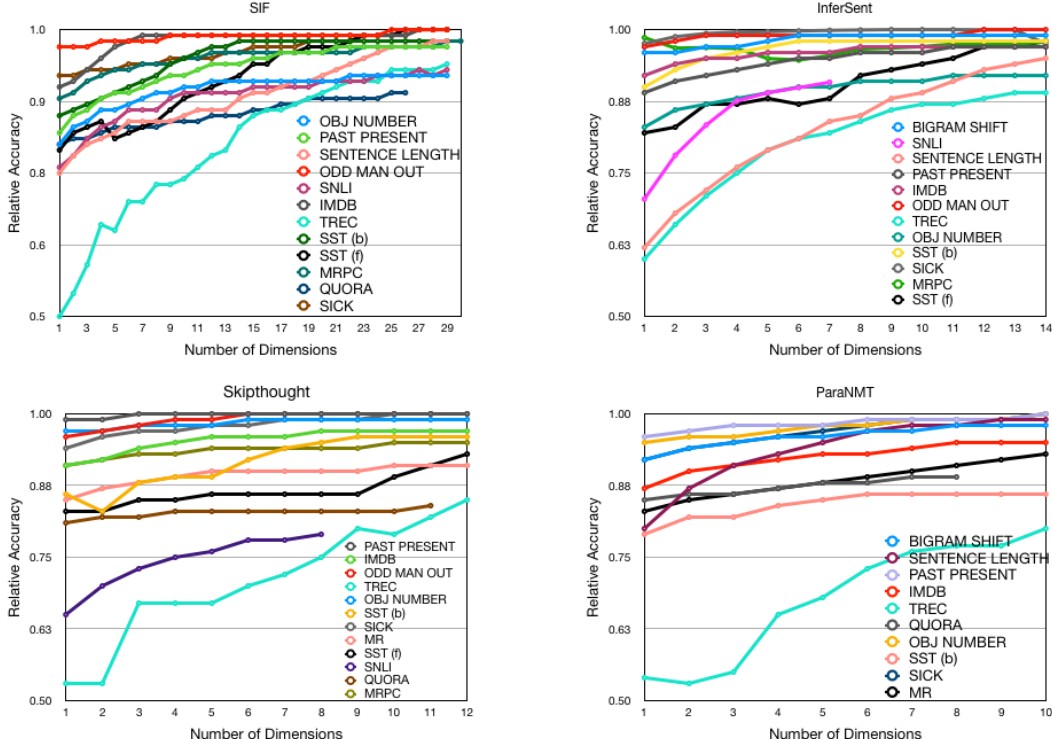

Figure 2: The *relative* accuracies of different tasks with increasing number of dimensions given by the greedy forward selection. Note that in majority of tasks, very few dimensions are required to reach close to the accuracy given by all the dimensions (i.e. 1.0 mark).

| **Validating Transfer Potential** (CFS / Clf. Weight Difference) | | | | | |
|---|---|---|---|---|---|
| **Task** | **Ranking Metric** | **InferSent** | **SkipThought** | **SIF** | **ParaNMT** |
| TREC | RR @1 | (0.25 / 0.20) | (0.50 / 0.33) | (0.25 / 0.33) | (0.50 / 0.33) |
| | Precision @5 | (0.60 / 0.80) | (0.80 / 0.60) | (0.40 / 0.80) | (0.80 / 0.60) |
| | NDCG @5 | (0.46 / 0.77) | (0.63 / 0.76) | (0.41 / 0.90) | (0.92 / 0.83) |
| SST (f) | RR @1 | (1.00 / 0.50) | (1.00 / 1.00) | (1.00 / 0.50) | (1.00 / 0.50) |
| | Precision @5 | (0.60 / 0.60) | (0.80 / 0.60) | (0.60 / 0.40) | (0.80 / 0.40) |
| | NDCG @5 | (0.85 / 0.61) | (0.92 / 0.99) | (0.68 / 0.60) | (1.00 / 0.60) |
| MRPC | RR @1 | (0.14 / -) | (0.20 / -) | (0.14 / -) | (0.17 / -) |
| | Precision @5 | (0.60 / -) | (0.80 / -) | (0.80 / -) | (0.60 / -) |
| | NDCG @5 | (0.63 / -) | (0.56 / -) | (0.71 / -) | (0.64 / -) |
| SICK | RR @1 | (1.00 / -) | (0.33 / -) | (0.33 / -) | (0.14 / -) |
| | Precision @5 | (0.80 / -) | (0.80 / -) | (0.60 / -) | (0.80 / -) |
| | NDCG @5 | (0.95 / -) | (0.55 / -) | (0.63 / -) | (0.16 / -) |

Table 2: Validating the transfer potential predicted by Condensed Feature Set (CFS) and Classifier (Clf.) Weight Difference. We evaluate our predicted ranked list against the gold utility transfer list using **NDCG @5**, which is Normalized Discounted Cumulative Gain, a measure of ranking quality biased towards top ranks. **RR @ 1** or Reciprocal Rank, equals $1/R$ where $R$ is the position in the predicted list where the top item of gold list appears. The Clf. weight difference method is not applicable for 2 sentence tasks (as discussed in §6)

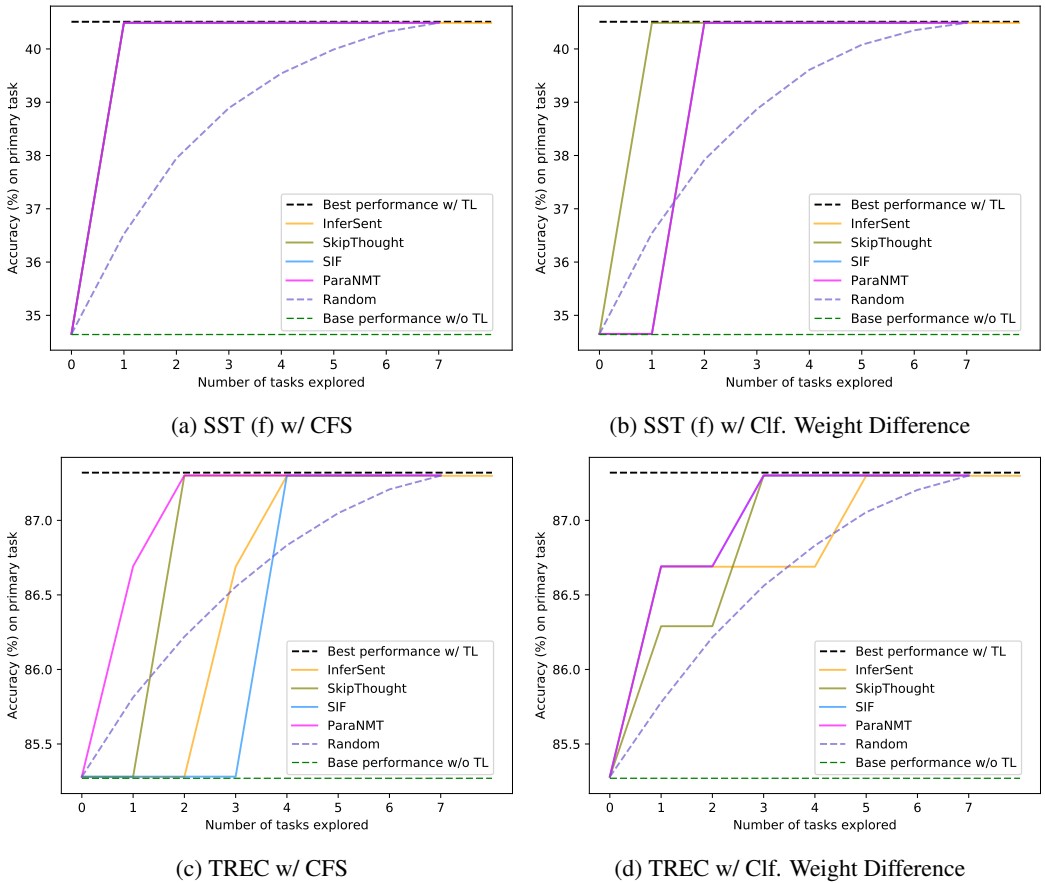

(a) SST (f) w/ CFS

(b) SST (f) w/ Clf. Weight Difference

(c) TREC w/ CFS

(d) TREC w/ Clf. Weight Difference

Figure 3: A graphical view depicting the transfer-learning (TL) gains for two tasks (`SST(f)` and `TREC`). These tasks are trained using pre-trained representations of other tasks, in the order of predictions from our two methods (`CFS` and Clf. Weight Difference). Each figure shows the maximum TL gain (y-axis) by transfering from the *best* task explored until that point (x-axis). For comparison, a random baseline gives expected best-TL gain for a random ordering of tasks.

## 7.2 RESULTS ON INFORMATION SIMILARITY AND APPLICATION TO TRANSFER LEARNING

Further, we determine the transfer potential among different tasks by using (a) transfer through the condensed feature sets and (b) similarity of classifier weights trained on different tasks. To evaluate the predictions about transfer potential by our proposed methods, we obtain "gold" transfer learning results for 4 low resource tasks – `TREC`, `MRPC`, `SST(f)` and `SICK`, where we train neural models for these tasks, without transfer, as well as, with transfer from each of the potential candidate tasks. We observe that for `SST(f)`, the biggest gain is by pretraining on `SST(b)`, `IMDB` in the given order, which happen to be movie review tasks just like `SST(f)`. Further, for `SICK`, the best tasks to transfer from are `SNLI`, `QUORA` which are related tasks of textual entailment and question deduplication respectively. The details of the actual transfer results are tabulated for all four primary tasks in Table 1 of Appendix.

To assess the utility of our methods' predictions about transfer potential, we compare our recommended list and the ranked gold list using metrics to compare rankings (like Precision @5, NDCG @5, Reciprocal Rank). From table 2, we can see that our top ranks are quite similar to the top gold ranks. The average Precision @5 by the CFS method is 70%, indicating that on an average 3.5 tasks out of top 5 tasks are predicted correctly. Whereas, the average Precision @ 5 for the classifier weight difference is 60%. Comparing CFS transfer and classifier weight difference head to head, we note that on 7 out of 8 occassions, CFS metric has a higher Reciprocal Rank @1, whereas on 5 out of 8 occasions it has a higher NDCG value as well.

Qualitatively, we predict that `SST(binary)` has the highest transfer potential for `SST(fine)` (unanimously predicted by all 4 sentence representations), while `SNLI` is predicted as the best candidate for 1 out of 4 sentence representations, and is in top 3 rankings of two other representations. These results match our intuition, and the gold transfer results. Figure 3 presents a graphical view of the utility of our predictions compared against the actual transfer results for `SST(f)`, and `TREC`. A similar graph for `SICK` and `MRPC` is Figure 1 in Appendix.

## 8 RELATED WORK

Below, we discuss some relevant research in understanding information in representations, transfer learning and finding overlap among tasks:

**Understanding Information in Representations:** Shi et al. (2016b) investigate if the sequence to sequence (seq2seq) models for machine translation learn syntactic information about the source sentence. They verify syntactic properties of the encoded representation of the source sentence. Similar to our goal, they study the extent and the concentration of information captured. They verify that their encoder captures to varying degrees sentence-level syntactic labels (like voice and tense of the sentence) and word-level syntactic labels (like part-of-speech tags of each word).

Another related work (Shi et al., 2016a) finds the encoded representations to be aware of the length of the source sentence, again for the task of machine translation. Karpathy et al. (2015) reveal the existence of cells that keep track of long-range dependencies such as line-lengths, quotes and brackets. Radford et al. (2017) observe, that a single dimension in the encoded representation controls the sentiment in the task of language modeling. This is remarkable and surprising, as one would not expect a high level concept like sentiment to be so clearly disentangled, especially for a task whose objective did not model sentiment explicitly.

**Transfer Learning:** Pre-training neural models using unsupervised data is used in wide variety of tasks such as using word embeddings for sentence classification (Kim, 2014) and pre-trained language models for Machine Translation (Ramachandran et al., 2016). More recently, general purpose sentence embeddings are used to solve downstream syntactic and semantic sentence level tasks. The sentence embeddings might be trained in a supervised or an unsupervised manner. For instance, SkipThought vectors (Kiros et al., 2015) tries to reconstruct the surrounding sentences, InferSent vectors (Conneau et al., 2017) are trained on the SNLI labeled corpus. These sentence embeddings have shown to perform well on downstream tasks such as sentiment analysis, paraphrase detection, entailment, etc, and hence, they must capture syntax and semantics of a sentence.

**Task overlap:** In a transfer learning setting for neural models, we wish to answer the question: which tasks can be used as auxiliary tasks or grouped together? Few past works try to answer this question in the context of multi-task learning. Kumar & Daume III (2012) assume that the vector formed using parameters of each task is a linear combination of a finite number of underlying 'basis' tasks. Then they use sparse matrix factorization to cluster tasks into similar tasks. Another approach can be to decorrelate the features explicitly while training. As long as the inputs for different tasks share the same representation, simple set overlap (discussed in §4.1) between the features might be enough to conclude high similarity. Cogswell et al. (2015) employ a regularization term to decorrelate activations in the last layer to achieve this. In the problem of domain adaptation, Stacked Denoising Auto-encoders (Yang & Eisenstein, 2014) are used to project the inputs into a high variance non-linear space, but they don't consider the labels. All these approaches are computationally intractable as they require training of multiple neural models, while our framework select tasks that requires training only simple classifiers.

## 9 CONCLUSION

In this work, we gauge information density in the hidden layers of neural networks trained for many natural language tasks. For most of the tasks we investigate, we find that the majority of the information is captured by merely a few units. Further, we apply these insights to transfer learning and present a framework to determine the transfer potential across different tasks in a computationally efficient way. We evaluate our predictions about transferability against the actual transfer learning performance and find them to be largely consistent.

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

# Appendix: Measuring Density and Similarity of Task Relevant Information in Neural Representations

**Anonymous authors**

## 1 Actual Transfer Results

| Main Tasks | TREC | MRPC | SICK | SST (f) |
|:---:|:---:|:---:|:---:|:---:|
| w/o transfer | 85.28 | 66.53 | 60.63 | 34.65 |
| SST (b) | 87.30 | 66.59 | 60.20 | 40.49 |
| IMDB | 84.48 | 66.82 | 59.65 | 39.22 |
| SNLI | 85.08 | 67.93 | 61.26 | 37.14 |
| QUORA | 83.47 | 67.64 | 60.97 | 33.61 |
| BShift | 77.82 | 67.87 | 56.90 | 31.30 |
| Tense | 83.06 | 68.12 | 59.79 | 33.56 |
| CoordInv | 83.67 | 68.75 | 60.16 | 33.24 |
| SOMO | 86.29 | 68.46 | 59.81 | 36.68 |
| SentLen | 86.69 | 67.54 | 57.13 | 25.54 |

Table 1: Exact Transfer Results: Accuracies obtained on four tasks (`TREC`, `MRPC`, `SICK`, `SST(f)`) by pre-training on all the other tasks. The top row (w/o transfer) denotes results without any transfer learning, and below we enlist the results with transfer learning.

## 2 PREDICTED RANKINGS OF TASKS

| | Predictions (using CFS) | | | |
|---|---|---|---|---|
| Ranks | **InferSent** | **SkipThought** | **SIF** | **ParaNMT** |
| | TREC | | | |
| 1 | SNLI | SNLI | SNLI | SentLen |
| 2 | BShift | SST (b) | QUORA | SST (b) |
| 3 | SentLen | IMDB | Tense | Tense |
| | MRPC | | | |
| 1 | SentLen | SST (b) | SNLI | SentLen |
| 2 | QUORA | IMDB | QUORA | BShift |
| 3 | SST (b) | SNLI | SST (b) | QUORA |
| | SICK | | | |
| 1 | SNLI | SST (b) | IMDB | SST (b) |
| 2 | SOMO | IMDB | QUORA | IMDB |
| 3 | QUORA | SNLI | SNLI | BShift |
| | SST (f) | | | |
| 1 | SST (b) | SST (b) | SST (b) | SST (b) |
| 2 | BShift | IMDB | SNLI | IMDB |
| 3 | IMDB | SNLI | QUORA | SNLI |

Table 2: Top 3 predictions of tasks using CFS in the order of transfer potential

# 3 TRANSFER EVALUATION

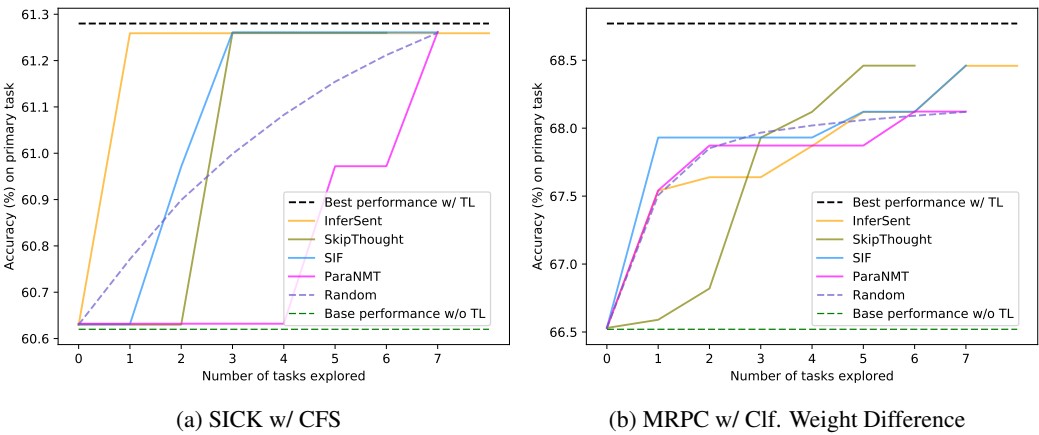

(a) SICK w/ CFS                    (b) MRPC w/ Clf. Weight Difference

Figure 1: A graphical view depicting the transfer-learning (TL) gains for two tasks (`SICK` and `MRPC`). These tasks are trained using pre-trained representations of other tasks, in the order of predictions from CFS information transfer. Each figure shows the maximum TL gain (y-axis) by transfering from the *best* task explored until that point (x-axis). For comparison, a random baseline gives expected best-TL gain for a random ordering of tasks.

# 4 INDIVIDUAL TASK, REPRESENTATION PLOTS

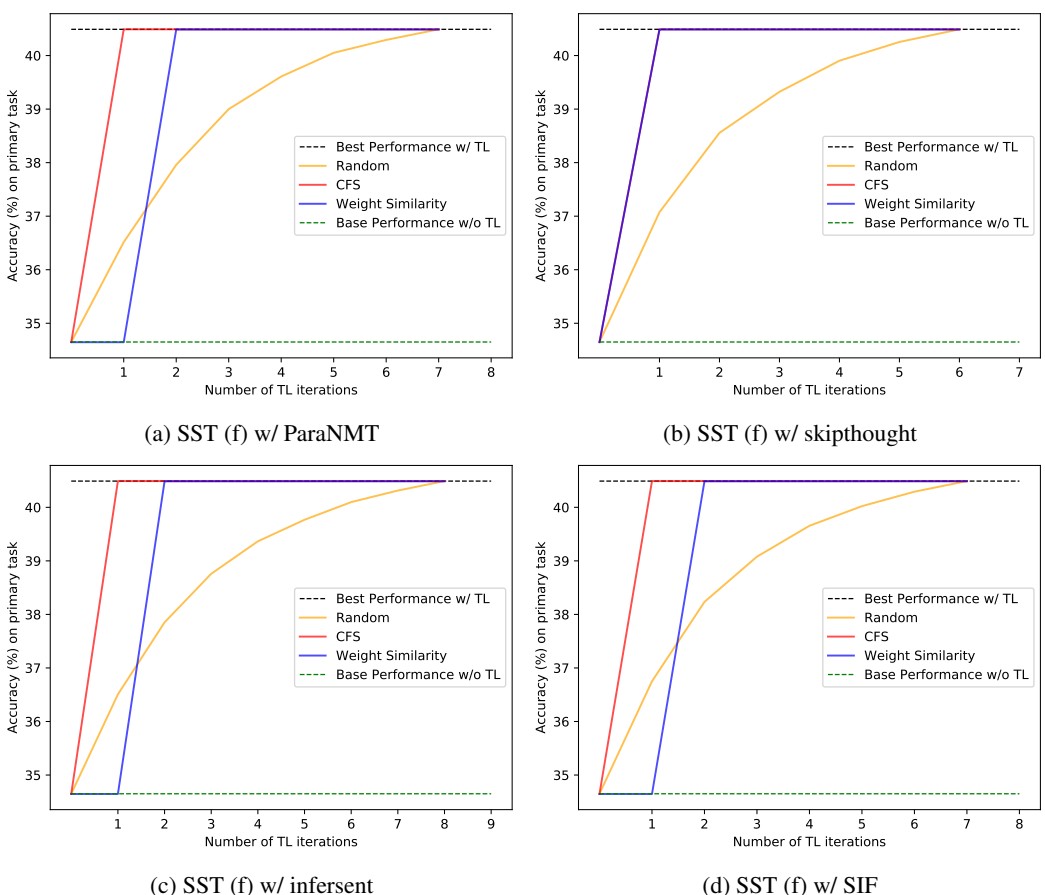

(a) SST (f) w/ ParaNMT

(b) SST (f) w/ skipthought

(c) SST (f) w/ infersent

(d) SST (f) w/ SIF

Figure 2: SST (f) plots for different input representations.

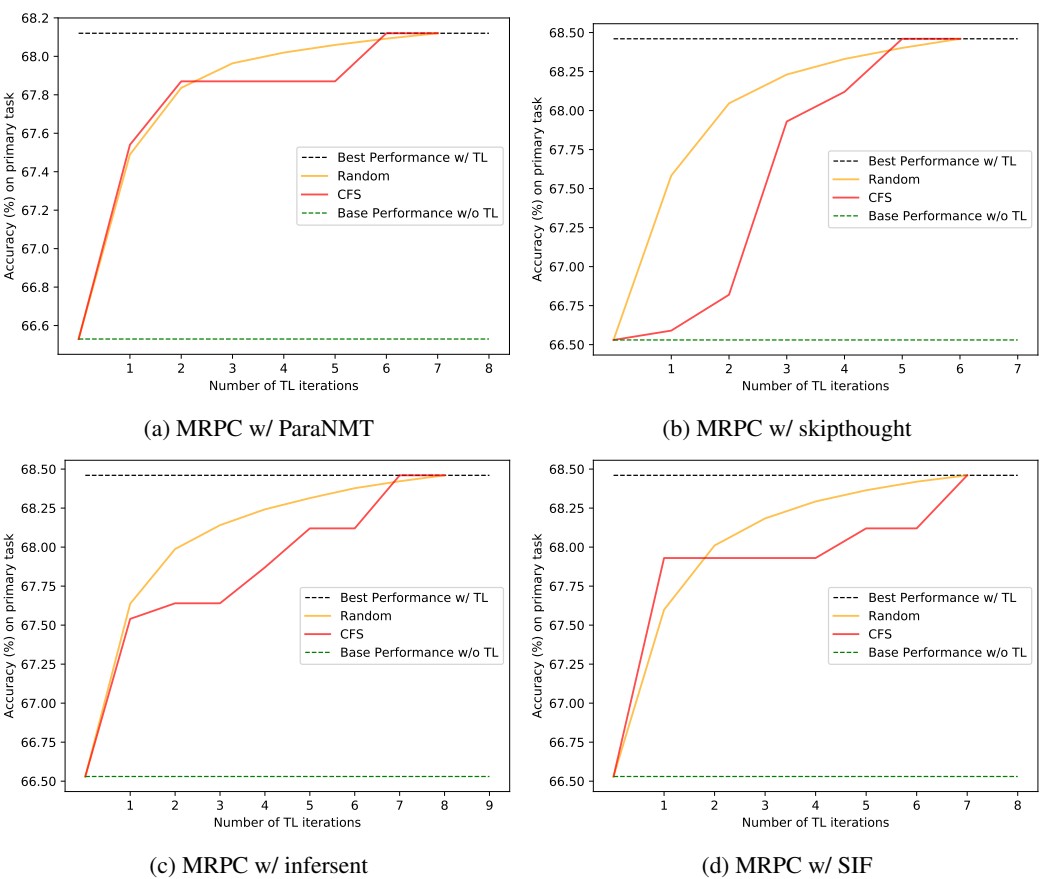

(a) MRPC w/ ParaNMT

(b) MRPC w/ skipthought

(c) MRPC w/ infersent

(d) MRPC w/ SIF

Figure 3: MRPC plots for different input representations.

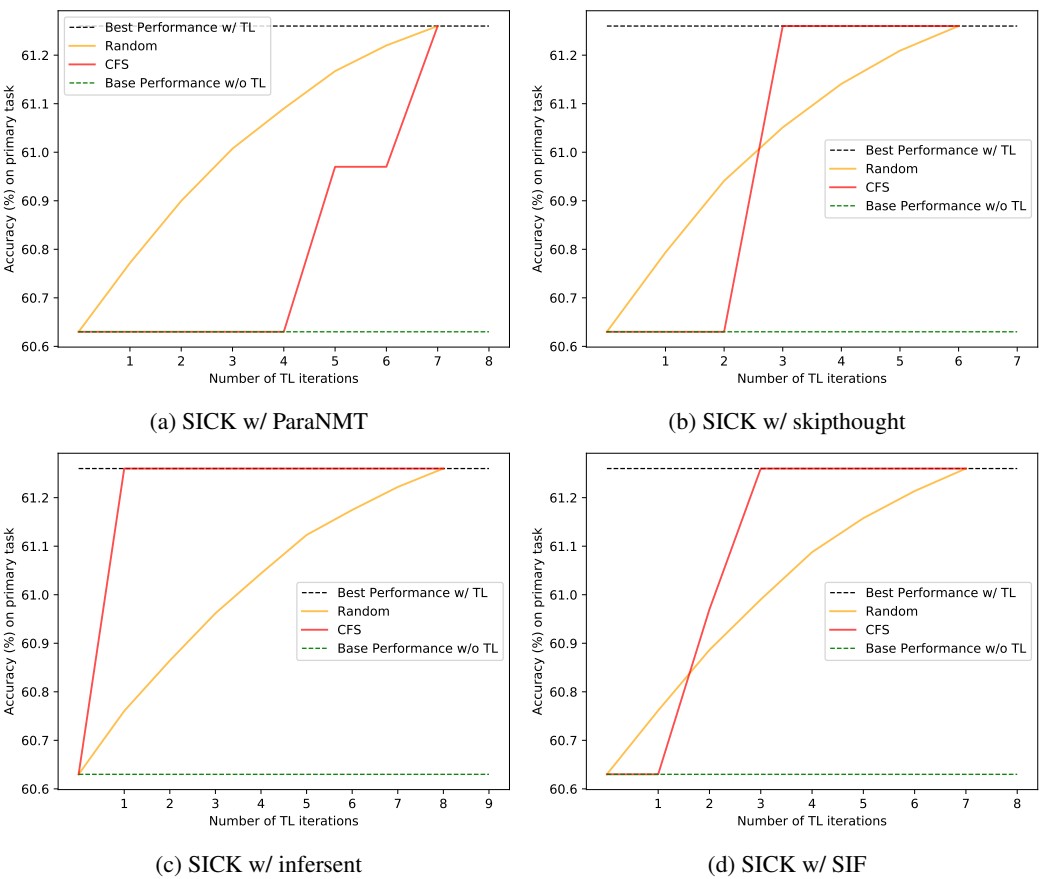

(a) SICK w/ ParaNMT

(b) SICK w/ skipthought

(c) SICK w/ infersent

(d) SICK w/ SIF

Figure 4: SICK plots for different input representations.

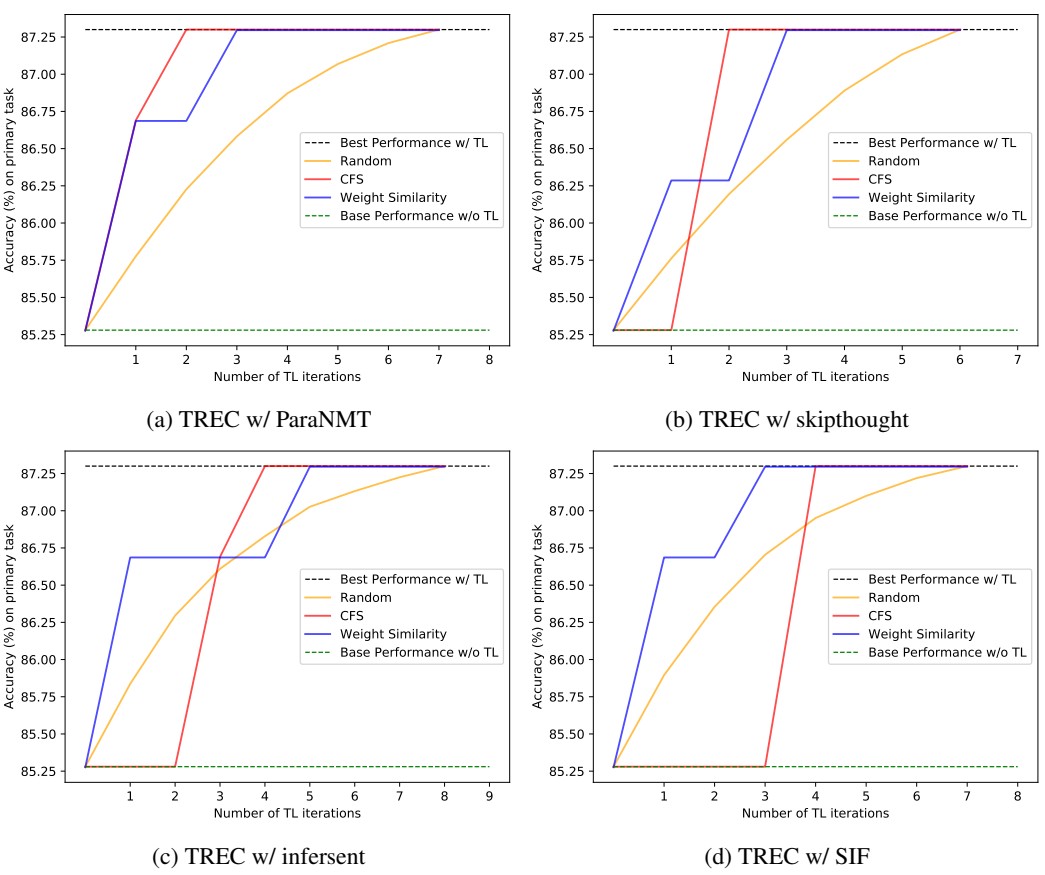

(a) TREC w/ ParaNMT

(b) TREC w/ skipthought

(c) TREC w/ infersent

(d) TREC w/ SIF

Figure 5: TREC plots for different input representations.

