# OpenReview forum: "Measuring Density and Similarity of Task Relevant Information in Neural Representations"
_ICLR.cc/2019/Conference_

### Official Review · AnonReviewer2 · 2018-11-01
**An interesting direction, but limited apparent utility**

**Rating:** 5
**Confidence:** 3

**Review:**

This paper proposes simple metrics for measuring the "information density" in learned representations. Overall, this is an interesting direction. However there are a few key weaknesses in my view, not least that the practical utility of these metrics is not obvious, since they require supervision in the target domain. And while there is an argument to be made for the inherent interestingness of exploring these questions, this angle would be more compelling if multiple encoder architectures were explored and compared.

+ The overarching questions that the authors set out to answer: How task-specific information is stored and to what extent this transfers, is inherently interesting and important.

+ The proposed metrics and simple and intuitive.

+ It is interesting that a few units seem to capture most task specific information.

- The envisioned scenario (and hence utility) of these metrics is a bit unclear to me here. As noted by the authors, transfer is most attractive in low-supervision regimes, w.r.t. the target task. Yet the metrics proposed depend on supervision in the target domain. If we already have this, then -- as the authors themselves note -- it is trivial to simply try out different source datasets empirically on a target dev set. It is argued that this is an issue because it requires training 2n networks, where n is the number of source tasks. I am unconvinced that one frequently enough has access to a sufficiently large set of candidate source tasks for this to be a real practical issue.

- The metrics are tightly coupled to the encoder used, and no exploration of encoder architectures is performed. The LSTM architecture used is reasonable, but it would be nice to see how much results change (if at all) with alternative architectures.

- The CFS metric depends on a hyperparameter (the "retention ratio"), which here is arbitrarily set to 80% without any justification.

- What is the motivation for the restriction to linear models? In the referenced probing paper, for example, MLPs were also used to explore whether attributes were coded for 'non-linearly'.

---

> ### Author Response · Authors · 2018-11-23
> **Thank you for your review. Please find a few clarifications below.**
>
> Thank you for your feedback. We are glad to know that you find the problem inherently interesting and important.
>
> Re: no exploration of encoder architectures is performed
> > We are not sure if we understand this completely. Just to clarify, we do compare 4 different sentence encoders [1][2][3][4] which display a fair amount of variety in ways which sentence representations can be computed. For instance, SkipThought vectors [1] use bi-GRU based encoder-decoder model to reconstruct the surrounding sentences. ParaNMT [2] and InferSent [3] use different LSTM based architectures to perform back-translation and textual entailment respectively. Lastly, SIF [4] is a tf-idf based weighted average of individual GloVe word representations.
>
> One of the key findings of our work is that task-specific information is captured succinctly for a majority of 13 different NLP tasks across 4 different choices of encoder architectures.
>
> 1. Skip-Thought Vectors (https://arxiv.org/pdf/1506.06726.pdf)
> 2. PARANMT-50M: Pushing the Limits of Paraphrastic Sentence Embeddings with Millions of Machine Translations (https://arxiv.org/pdf/1711.05732.pdf)
> 3. Supervised Learning of Universal Sentence Representations from Natural Language Inference Data (https://arxiv.org/pdf/1705.02364.pdf)
> 4. A Simple but Tough-to-Beat Baseline for Sentence Embeddings (https://openreview.net/forum?id=SyK00v5xx)
>
>
> Re: Utility of the methods is a bit unclear
> > We agree that our approach to estimate transfer potential reaps true benefits only when n is large. However, this is not uncommon in scenarios like machine translation, where there are hundreds of potential language pairs that could be used as candidate tasks.
>
> Furthermore, we believe (although acknowledge that this is subjective) that curiosity-driven questions about how the information is encoded are interesting: while they might not be useful in a way that is easily measurable by quantifiable metrics, they provide insights that can help guide future work.
>
>
> Re: CFS metric depends on a hyperparameter (the "retention ratio")
> > Sorry about the lack of clarity! To clarify, we used the elbow method (used to find an appropriate number of clusters for clustering) and observed that the ‘elbow’ in the accuracy vs dimensions plot was around the 80% accuracy mark for most tasks, and hence, we used 80% as the retention ratio. We will discuss this process and test with different retention ratios in the final version.
>
>
> Re: motivation for the restriction to linear models?
> > Our motivation to use linear models is to keep the setup simple and fast. As the classifiers are able to extract task-specific information and reliably estimate transfer potential; changing to a different classifier like MLP, we believe, shouldn’t affect our results in a significant way. However, we will empirically verify this, and discuss this in the camera-ready/future versions of the paper.

---

### Official Review · AnonReviewer1 · 2018-11-02
**An interesting approach to an important problem; but limited in scope and relevant comparisons**

**Rating:** 5
**Confidence:** 4

**Review:**

MEASURING DENSITY AND SIMILARITY OF TASK RELEVANT INFORMATION IN NEURAL REPRESENTATIONS

Summary:

This work attempts to define two kinds of metrics (metrics for information density and for information similarity) for the sake of automatically detecting similarity between tasks so that transfer learning can be done more efficiently. The concepts are clearly explained, and the metric for information density seems to match up with intuitions coming out of forward selections approaches. The metric for information transfer seems to be the commonplace metric that other works default to when they show that pre-trained representations are effective on downstream tasks. It is not clear that the notion of similarity through classifier weights makes sense, but see below for clarification questions. The problem addressed (automatic similarity scoring of tasks) is important for transfer learning, and thus the results have potential to be very impactful if they generalize to other kinds of tasks; as is, they seem to apply only to classification tasks, but that is a good step.

Pros:

Clearly written; experiments on the datasets chosen do seem to suggest that the proposed methods have potential. Brings in nice intuition from forward feature selection. An important problem with potential for high impact.


Cons:

It is not clear to me that the classifier difference metric is well-defined. Is there a constraint on the CFS and classifiers that ensure the difference between the weights really captures what is suggested? Is it not the case that classifier weights could come out quite different despite the tasks being quite similar if the linear classifiers learned to capitalize on dissimilar, yet equally fruitful patterns in the input features?

Do you have thoughts on how this could be applied outside the context of sentence representations and further outside the context of classification? Those seem to be quite limiting features of these methods, which is not to say that they are not useful in that realm, but only to clarify my understanding of their possible scope of application.

These classification datasets are often so close, that I do wonder whether even simpler methods would work just as well. For example, clustering on bags-of-words might also show that SST, SST-fine, and IMDb are close/similar/transferable. The same could be said for SICK and SNLI. It would be nice to see a comparison to such baselines in order to get a sense of how the proposed methods give insights that other unsupervised or supervised methods might give just as well. Otherwise, it is hard to tell how significant these correlations are. Since the end goal is to determine transferability of tasks and not the methods, it does seem like there are simpler baselines that you could compare against.

---

> ### Author Response · Authors · 2018-11-23
> **Thanks for the review! A few clarifications below**
>
> We thank you for your thoughtful review. We are happy to learn that you believe it is an interesting direction that holds potential for high impact.
>
> Re: simpler methods (like clustering, BoW etc.) might work equally well
> > To assess transfer learning potential reliably, we require both the X and y for the target task (i.e supervision). Consider the case where the target task is sentiment analysis, and one of the candidate tasks is finding sentence length (SentLen). For the sake of the argument, let us assume that the X for both sentiment analysis and sentence length is exactly the same set of movie reviews. In such a case, unsupervised metrics like clustering, BoW etc. would indicate maximum transfer potential, whereas the actual transfer potential would be close to zero (assuming the lengths of reviews aren’t correlated with the sentiment). This is a fundamental problem of measures that look directly at the input data X without considering the nature of the labels y.
> For the sake of completeness, we will compare our methods with the suggested baselines in the camera ready/future versions of the paper.
>
>
> Re: not clear if the classifier weight difference is well defined
> > You are right in noting that the classifier weights might capture dissimilar yet useful features for two similar tasks, and hence the classifier weight difference might under-predict the transfer potential. We discuss this issue in the paper (section 4.1), which is why we avoid the set overlap metric. Owing to similar concerns, we recommend using CFS information transfer metric over classifier weight difference (which is also supported by results in Table 2 and Figure 3).
>
>
> Re: thoughts on how this could be applied outside the context of sentence representations and classification
> > It is easy to adopt our approach to study the information encoded in the encoders for other problems involving structured prediction (say POS Tagging). Instead of using a decoder that takes in all the dimensions of the encoded input token, one could iteratively select dimensions that provide the highest gains in decoding the right target sequence (say POS tags). Our formulation is very general, and it could potentially also be applied to other modalities like images for tasks like image classification and captioning.

---

> > ### Comment · AnonReviewer1 · 2018-11-26
> > **Thank you for elaborating and clarifying**
> >
> > I appreciate the time you took to explain your reasoning about simpler methods, and I look forward to the comparisons you mentioned.
> >
> > It also does sound like you you've already thought about how to adapt these ideas to other settings, which I think will be a good next test for these methods.

---

### Official Review · AnonReviewer3 · 2018-11-07
**Some nice pieces and ideas but have concerns about methodology and shallow analysis**

**Rating:** 4
**Confidence:** 4

**Review:**

This paper tries to quantify how "dense" representations we need for a specific task -- more specifically, how many dimensions are needed from a given representation (for a given task) to achieve a percentage of the performance of the entire representation. The second thing the paper tries to quantify is how well representations learned for one task can be fine tuned for another. Experiments are conducted with 4 different representation technique on a dozen or so tasks.

Quick summary: While I liked aspects of this -- including the motivation of having a lightweight way of understanding how well representations transfer across tasks, overall my concerns surrounding the methodology and some missing analysis leads me to believe this needs more work before it is ready for publication.

Quality: Below average
I believe the proposed techniques have some flaws which hurt the eventual method. There are also concerns about the motivations behind parts of the technique.

Clarity: Fair
There were some experimental details that were poorly explained but in general the paper was readable.

Originality: Fair
There were some nice ideas in the work but I remain concerned about aspects of it.

Significance: Below average
My concern is that the flaws in the method do not make it conducive to use as is.


Strengths / Things I liked:

+ I really liked the motivating problem of being able to (hopefully cheaply / efficiently) estimate transfer potential to understand how well representations will perform on a different task.

+ Multiple representations and tasks experimented with

Weaknesses / Things that concerned me:
(In no specific order)

- (W1) Adversely affected by rotations: One of my big concerns with the work is the way the CFS is computed. While it seems ok to estimate these different metrics using only linear models, my concern with this is that the linear models are only given a subset of the **exact** dimensions of the original representations. This is very much unlike the learning objectives of most of these representation learning methods and hence is highly biased and dependent on the actual methods and the random seeds used and the rotations it performs. (In many cases the representations are used starting with a fully connected layer bottom layer on top of the representations and hence rotations of the representations do not affect performance)

Let's take an example: Say there is a single dimension of the representation that is a perfect predictor of a task. Suppose we rotated these representations. Now the signal from the original dimension is split across multiple dimensions and hence the CFS may be deceivingly high.

To me this is a big concern as different runs of the same representation technique can likely have very different CFS scores based on initializations and random seeds.

- (W2) Related to the last line: I did not see any experiments / analysis showing how stable these different numbers are across different runs of the representation technique. Nor did I see any error bars in the experiments. This again greatly concerned me as I am not certain how stable these metrics are.

- (W3) Baselines for transfer learning: I felt this was another notable oversight. I would have liked to see results for both trivial baselines like random ranking as well as more informed baselines where we can estimate transfer potential using say k representation techniques, and then use that to help us understand how well it would do on the other representations. This latter baseline is a zero-cost baseline as it is not even dependent on the method.

- (W4) Metrics for ranking of transfer don't make sense (and some are missing) : I also don't understand how "precision" and NDCG are used as metrics. Based on my understanding the authors rank (which itself is questionable) the different tasks in order of potential for transfer and then call this the "gold" set. How is precision and NDCG calculated from this?

More importantly I don't believe looking at rank alone is sufficient since that completely obscures the actual performance numbers obtained via transfer. In most cases I would care about how well my model would perform on transfer not just which tasks I should transfer from. I would have wanted to understand something like the correlation of these produced scores with the actual ground truth performance numbers.

- (W5) Multi-task learning: I did not see any mention or experiments of what can be expected when the representations are themselves trained on multiple tasks. (This seems like something that could easily be done in the empirical analysis as well and would provide richer empirical signals as well)

- (W6) Motivation for CFS: I still don't fully understand the need to understand the density of the representation (especially in the manner proposed in the paper). Why is this an important problem? Perhaps expanding on this would be helpful

- (W7) Alternatives to CFS / Computational concerns: A big concern I had was the computational expense of the proposed approach. Unfortunately I did not see any discussion about this in the paper or empirically.

I find this striking because I can easily come up with cheaper alternatives to get at this "density". For example using LASSO / LARS like methods you can perhaps figure out a good reduced dimension set more efficiently.

If I were to go through the computation of then why not just train a smaller version of that representation technique instead and **directly** see how well it can encode data in k dimensions via that technique / for that task?

Alternatively why not try using a factorization technique to reduce the rank and then see how well the method does for different ranks?

- (W7b) Likewise I wonder if we could just measure transfer more directly as well and why we need to go via these CFS sets

- (W8) The proposed  CLF weight difference method has some concerning aspects as well. For example say we had two task with exact opposite labels. They would have a very low weight difference score though they are ideal representations for each other. Likewise looking at a difference of weight vectors seems arbitrary in other ways as well.

---

> ### Author Response · Authors · 2018-11-23
> **Thanks for the review! A few clarifications below**
>
> We thank the reviewer for their insightful and constructive feedback.
>
> Re: (W1 & W2) Adversely affected by rotations
> > While the CFS is not invariant to rotations, it is a surprising, and empirically noteworthy, finding that all 4 different ways of producing representations consistently encode a dozen tasks very succinctly. This is in line with some early work that observed that certain characteristic properties like length [1][2], sentiment [3], presence/absence of brackets [4] are encoded in a single dimension in the space.
>
> Some of these findings can be attributed to the additive property of the cell state of the LSTMs c_t = f_t c_{t-1} + h_t {c’}_{t}, which is free from matrix rotations. As previously noted, this empirically also results in single cells of the LSTM being interpretable. To just give one example, LSTM cell state can increment by a fixed amount at every time step and can count the number of tokens reliably (i.e the string length) [1][4].
>
>
> Re: (W3) Baselines for transfer learning:
> > The random baseline (i.e a random ordering of candidate task) is compared in figure 3 (and all the plots in the appendix), where we plot the accuracy boost using the best task till now in the produced recommendation of candidate tasks using different methods. We can clearly see that the random ordering is much worse compared to informed metrics that use representations. Upon your suggestion, we would also add this random baseline in table 2 as well.
>
>
>
> Re: (W4) Metrics for ranking of transfer don't make sense (and some are missing). How is precision and NDCG calculated
> > To compute the gold set, we first train a neural network for each of the candidate tasks and then use the pre-trained sentence encoder (part of the network) from the candidate task to fine-tune on the target task. The ranked list (in the decreasing utility of transfer learning gain) is then considered the ‘gold’ set.
>
> We further compare our recommendations of candidate tasks generated using CFS and classifier weight difference methods against the gold ranked list. Precision@K, Reciprocal Rank and NDCG are among the popular information retrieval metrics to compare a recommended list against a gold ranked list.
>
> These metrics are meaningful in our case, for instance, Reciprocal Rank tells us how many tasks we need to consider as per our recommendation before we hit the highest performing candidate task. Figure 3 presents the accuracy boost using the best task till now in the produced recommendations for the candidate tasks using different methods.
>
> Regarding missing values:
> As we explain in the paper, classifier weight difference metric is only applicable in cases
> where the number of features between the tasks are of the same size. Thus, 2 sentence input tasks and 1 sentence input tasks cannot be compared using the metric.
>
>
>
> Re: (W5) Multi-task learning
> > Our goal is somewhat orthogonal to the multitask learning setting where all the tasks are jointly trained. We, instead, focus on how the task-specific information is present in popular sentence representations, and how this could be used to assess transfer potential among tasks.
>
>
>
> Re: (W6) Motivation for CFS
> > There is a rich literature concerning what information is captured in the representations. Further, there are a few initial works that show that certain characteristics like length [1][2], sentiment [3], presence and absence of tokens like brackets [4] are densely captured in a single dimension of the representation space. In a similar spirit, we wanted to quantitatively study this surprising phenomenon, and we were curious about how densely is information encoded in representations.
>
>
> Re: (W7) Alternatives to CFS / Computational concerns
> > We agree that LARS/LASSO could act as potential ways to attain reduced dimensions. For our use case, we found the greedy forward selection computationally fast enough (of the order of a few minutes), and we observed a significant portion of accuracy captured in a very few dimensions. We would definitely explore this further, and add a detailed analysis on computational efficiency of different methods to reduce dimensions.
>
>
> Re: (W8) The proposed  CLF weight difference method has some concerning aspects as well. For example say we had two task with exact opposite labels. They would have a very low weight difference score though they are ideal representations for each other
> > You are right. For the very same reason, we take the inverse of the difference of normalized absolute classifier weights (Section 4.2).
>
> References:
> 1.“Why Neural Translations are the Right Length” :http://www.aclweb.org/anthology/D16-1248.pdf
> 2. On the Practical Computational Power of Finite Precision RNNs for Language Recognition: https://arxiv.org/abs/1805.04908
> 3. Learning to Generate Reviews and Discovering Sentiment: https://arxiv.org/abs/1704.01444
> 4. Visualizing and Understanding Recurrent Networks : https://arxiv.org/abs/1506.02078

---

### Public Comment · (anonymous) · 2018-11-07
**Now unneeded emergency fill-in review**

This is an emergency fill-in review that was originally asked for but now is unnecessary as the missing review was posted. Here it is anyways.

Review:

This paper attempts to answer two questions: how densely is information included in sentence representations and how similar are encodings from encoders learned from different tasks?

Pros:
	1) This paper analyzes representation from a perspective that seems distinct from previous work. It is somewhat in-line with work stating that NNs are heavily overparameterized, and this work might be considered how overparameterized the representations are for NLP tasks.
	2) They present a fairly new method for trying to predict what tasks might be useful pretraining for other tasks.
	3) Their motivation, thought process, and formalism for their method is well-written and very clear, if almost too long.

Cons:
Viewing this paper as making a methods contribution, I think the proposed approach is somewhat limited:
	1) the method is only applicable to linear classifiers. I understand practically the decision to use only linear classifiers, but this decision limits the set of representations that can be fairly studied with this method to only representations just before the final linear layer, as using other parts of the model's internal representation are confounded by the fact that they are optimized for use in non-linear models.
	2) the method is not comparable across tasks with the different input/output format (slightly mitigated by the fact that you can recast tasks, but it's hard to overcome the fundamental limitation of one input vs two input tasks without introducing some weirdness)
	3) the method seems limited to sentence-to-vector models
Also, it'd be nice to give the upper bound on the quality of the approximation for the proposed greedy algorithm (I imagine it's something like (1 - 1/e) and the runtime.

As an analysis paper, which I think is more compelling than as a methods paper, the results are fairly interesting, but there isn't enough discussion of the results and I have some concerns regarding the experiments:
	4) it would have been nice to have more quick experiments to sanity check the method for predicting transfer learning. Using the predicted transfer between SST2 and SST5 is a good starting point, but there could, and I think should, have been more, e.g.: between random subsets of the same task, between different genres within MNLI, between MNLI and SNLI.
	5) without a description of how the transfer learning is done, it's really hard to say how accurate these "gold" rankings of transfer learning are or what confounders are potentially introduced in their transfer learning approach
	6) I think there needed to be some explanation, even just a one sentence explanation of how hyperparameters were chosen, especially the \alpha parameter. How quickly does the algorithm pick the entire set as \alpha approach 1?
The discussion of results is very short relative to the density of the experiments and plots. The early exposition explaining everything mathematically and intuitively is nice, but I think the notation was somewhat superfluous and could have been condensed to include more analysis/discussion of the results.


Style / Presentation
	1) It'd be nice if each task was the same color across plots in Figure 2
	2) typos: section 3, p2: "...we first define accuracy score of the best classifier..."; section 1, last p: "...transferring the knowledge acquired therefrom to improve performance..."
	3) There's some related work analyzing contextual representations (outside sentence-to-vector) that would be worthwhile to mention, e.g. http://aclweb.org/anthology/D18-1179


Rating: 5
Confidence: 4

---

> ### Comment · Area_Chair1 · 2018-11-17
> **Thanks for posting this anyway!**
>
> - Your AC

---

> ### Author Response · Authors · 2018-11-23
> **Thanks for the review! A few clarifications below.**
>
> We thank the reviewer for the detailed and thorough reviews (that too, likely, on a short notice). We wish to clarify the following:
>
> Re: The method is only applicable to linear classifiers.
> > While the motivation for using linear classifiers was to keep our approach simple and computationally fast, our method can easily be extended to non-linear classifiers, by extending the set of functions (\mathcal{F}) in section 3 to non-linear functions. We will also, thus, verify the application of this method to the intermediate layers of the networks and will include our results in the camera-ready/future versions of this paper.
>
> Re: more quick experiments to sanity check the method for predicting transfer learning
> > This is a very good idea! We would further experiment on different splits of the data to sanity check our observations and key findings, and include the analysis in the final version of the paper.
>
> Re: description about the “gold” rankings for transfer learning
> > Sorry for the lack of clarity! To compute the “gold” transfer gains, we first train a neural network for each of the candidate tasks and then use the pre-trained sentence encoder (part of the network) from the candidate task to fine-tune on the target task. The candidate tasks are then ranked based on the improvements from this pretraining to compute the “gold” rankings.
>
> Re: explanation of how hyperparameters were chosen, especially the \alpha parameter
>
> > We discuss the motivation for the selection of \alpha parameter in section 7.1; sorry for not mentioning it clearly. To determine the parameter, we used the elbow method (used to find an appropriate number of clusters for clustering) and observed that the ‘elbow’ in the relative accuracy vs dimensions plot was around the 80% accuracy mark for most tasks (which can be inferred from in Figure 2).

---

### Meta-Review · Area_Chair1 · 2018-12-14
**Interesting direction, but no compelling new method yet**

**Confidence:** 4
**Recommendation:** Reject

**Metareview:**

This paper addresses important general questions about how linear classifiers use features, and about the transferability of those features across tasks. The paper presents a specific new analysis method, and demonstrates it on a family of NLP tasks.

All four reviewers (counting the emergency fourth review) found the general direction of research to be interesting and worthwhile, but all four shared several serious concerns about the impact and soundness of the proposed method.

The impact concerns mostly dealt with the observation that the method is specific to linear classifiers, and that it's only applicable to tasks for which a substantial amount of training data is available.

As the AC, I'm willing to accept that it should still be possible to conduct an informative analysis under these conditions, but I'm more concerned about the soundness issues: The reviewers were not convinced that a method based on the counting of specific features was appropriate for the proposed setting (due to rotation sensitivity, among other issues), and did not find that the experiments were sufficiently extensive to overcome these doubts.